# Targeting Hedgehog Pathway and DNA Methyltransferases in Uterine Leiomyosarcoma Cells

**DOI:** 10.3390/cells10010053

**Published:** 2020-12-31

**Authors:** Natalia Garcia, Ayman Al-Hendy, Edmund C. Baracat, Katia Candido Carvalho, Qiwei Yang

**Affiliations:** 1Department of Surgery, University of Illinois at Chicago, Chicago, IL 60607, USA; natalia.garciaft@gmail.com (N.G.); aalhendy@bsd.uchicago.edu (A.A.-H.); 2Laboratório de Ginecologia Estrutural e Molecular (LIM 58), Disciplina de Ginecologia, Departamento deObstetricia e Ginecologia, Hospital das Clinicas da Faculdade de Medicina da Universidade de Sao Paulo, HCFMUSP, SP, BR Av. Dr Arnaldo 455, sala 4121, Cerqueira Cesar, São Paulo 05403-010, Brazil; ecbaracat@gmail.com; 3Department of Obstetrics and Gynecology, University of Chicago, Chicago, IL 60637, USA

**Keywords:** uterine leiomyosarcoma, hedgehog signaling, inhibitor, DNA methyltransferases

## Abstract

Uterine leiomyosarcoma (LMS) is an aggressive tumor that presents a poor prognosis, high rates of recurrence, and metastasis. Because of its rarity, there is no information available concerning LMS molecular mechanisms of origin and development. Here, we assessed the expression profile of Hedgehog (HH) signaling pathway markers and the effects of their pharmacological inhibition on uterine smooth muscle (UTSM), leiomyoma, and LMS cells. Additionally, we also evaluated the effects of DNMTs inhibition on LMS cell behavior. Cell proliferation, migration and apoptosis rates were evaluated by MTT, Scratch, and Annexin V assays, respectively. RNA expression and protein levels were assessed by qRT-PCR and Western blot. We found that SMO and GLIs (1, 2, and 3) expression was upregulated in LMS cells, with increased nuclear levels of GLI proteins. Treatment with LDE225 (SMOi) and Gant61 (GLIi) resulted in a significant reduction in Glis protein levels in LMS (*p* < 0.05). Additionally, the expression of DNMT (1, 3a, and 3b), as well as GLI1 nuclear expression, was significantly decreased after treatment with HH inhibitor in LMS cells. Our results showed that blocking of SMO, GLI, and DNMTs is able to inhibit LMS proliferation, migration, and invasion. Importantly, the combination of those treatments exhibited a potentiated effect on LMS malignant features due to HH pathway deactivation.

## 1. Introduction

Uterine leiomyosarcoma (LMS) is a rare uterine cancer, representing 1–2% of all uterine malignancies [1]. The annual incidence of LMS is approximately 0.8 per 100,000 women [2]. The 5-year survival for all patients is between 25 and 76%, with survival for women with metastatic disease at the time of initial diagnosis approaching only 10–15% [3]. Irrespective of treatment, LMS is characterized by poor prognosis [4]; the present treatment for LMS patients exhibits resistance to currently available therapies, as evidenced by high rates of both recurrence and progression [5].

The first evidence of Hedgehog (HH) pathway deregulation in LMS patients was described by Garcia and collaborators [6]. Higher protein expression levels of SMO and GLI1 were found in LMS samples compared to myometrium and leiomyoma variants. Additionally, increased expression of SHH and SUFU was correlated with decreased overall survival [6].

The HH signaling pathway plays an essential role in several biological processes, including embryonic development, tissue differentiation, as well as the pathogenesis of multiple cancer types [6,7,8,9]. Activation of the canonical HH signaling pathway occurs when the HH ligand binds and inactivates PTCH1, releasing SMO protein signaling to its cytoplasm targets [10]. SMO is a G-protein-coupled, receptor-like (GPCR-like) protein, which triggers GLI proteins’ translocation to the nucleus and their consequent binding to DNA [11]. GLI1 has only an activator form, while GLI2 and GLI3 have both activators (GLI2A and GLI3A) and repressor forms (GLI2R and GLI3R) [12]. GLI members act as the nuclear effectors at the end of the pathway, which is responsible for regulating the expression of downstream target genes [13,14].

HH signaling pathway activation can also occur due to epigenetic mechanisms. It has been observed that the promoter region in HH ligand is hypomethylated in breast cancer cells [15,16]. GLI3 hypomethylation was observed in gastric cancer [17]. Moreover, the Hedgehog-interacting protein (HHIP) is silenced by promoter hypermethylation in lung and hepatocellular cancers, and this was correlated with the downregulation of the protein [18,19].

PTCH1, as a negative regulatory factor of the HH signaling pathway, could be involved in tumorigenesis. Hypermethylation at the PTCH1 promoter region has been described in rhabdomyosarcoma, medulloblastomas [20], and breast cancer [21,22], astrocytoma and medulloblastoma cell lines contributing to HH signaling activation [23,24]. In addition, alterations in the methylation status of *PTCH1* gene were reported in different types of cancer, suggesting the epigenetic role of *PTCH1* in tumor development [23,25,26,27]. In this study, we evaluated the potential targets of HH for anti-LMS therapy.

## 2. Materials and Methods

### 2.1. Cells and Reagents

The immortalized human leiomyoma cell line (HuLM) and immortalized human uterine smooth muscle (UTSM) cells were a generous gift from Professor Darlene Dixon. The cells were cultured and maintained in phenol red-free, 10% fetal bovine serum Dulbecco’s Modified Eagle Medium: Nutrient Mixture F-12. The leiomyosarcoma (LMS) cell line (SK-UT1, ATCC^®^ HTB-114^TM^) (ATCC, Manassas, VA, USA) was cultured and maintained in ATCC-formulated Eagle’s Minimum Essential Medium with 10% of fetal bovine serum. We used these three cell lines covering the spectrum from a normal cell line (UTSM), benign uterine tumor cell line (HuLM), and uterine malignant cell line (LMS) to better understand the tumor progression linking to the HH pathway.

SMO inhibitors LDE225 and GDC0449 were purchased from Selleck Chemical (Houston TX, USA), GLI inhibitors Gant58 and Gant61 from Sigma Aldrich (St. Louis, MO, USA) and DNA methylation inhibitor 5′ Aza- 2′-deoxycytidine from Biosynth & Carbosynth (Staad, St. Gallen, Switzerland). The range of doses tested was 0.1–60 µM.

### 2.2. Proliferation Assay

Cell proliferation was measured using dimethythiazoldiphenyltetra-zoliumbromide (MTT Sigma Aldrich, St. Louis, MO, USA) assay. A total of 2 × 10³ cells per well were seeded into 96-well tissue culture plates, treated as described in the figure legends with the SMO (GDC0449 and LDE225) and GLI (Gant61 and Gant58) inhibitors, and MTT assay was performed at different time points (24, 48, and 72 h). Absorbance was measured in a synergy HT multi-detection microplate reader (BioTek, Winooski, VT, USA) at 570 nm. This assay was performed three times in triplicate.

### 2.3. Cell Treatment Using Hedgehog Pathway and DNA Methyltransferase Inhibitors

LMS cells were seeded at 8 × 10^4^ per well in a six-well plate and cultured overnight, then LMS cells were treated with SMO- LDE225 (10 µM), GLI-Gant61 (30 µM), or DNMT- 5′-Aza-dc (2 µM) inhibitors for 72 h, with daily replacement/change. After the treatment, the cells were harvested for protein/RNA expression measurement and other studies. The experiments were performed three times in triplicate.

### 2.4. RNA Extraction and Gene Expression

Total RNA was isolated using Trizol reagent (Invitrogen, Carlsbad, CA, USA). The concentration of total RNA was determined using NanoDrop (Thermo Scientific, Waltham, MA, USA). One microgram of total RNA from each sample was reverse-transcribed to complementary DNA (cDNA) using the High Capacity cDNA Transcription Kit (Thermo Scientific, Waltham, MA, USA). Quantitative real-time polymerase chain reaction (qRT-PCR) was performed to determine the messenger RNA (mRNA) expression of several genes listed with their primer sequences in Appendix A; all primers were selected from the literature and the sequences were confirmed using Primer-Blast (https://www.ncbi.nlm.nih.gov/tools/primer-blast/). The real-time PCR reactions were performed using CFX96 PCR instrument using SYBR Green Supermix (Bio-Rad, Hercules, CA, USA). *GAPDH, B2M, 18S* and *β-ACTIN* were tested as housekeeping genes, and B2M was used as an internal control. The results are presented as relative gene expression using CFX Maestro^TM^. This assay was performed three times in triplicate.

### 2.5. Protein Extraction and Western Blot

Cells were collected and lysed in a RIPA lysis buffer with protease and phosphatase inhibitor cocktail (Thermo Scientific, Waltham, MA, USA), the protein was quantified using the Bradford method (Bio-Rad Protein Assay kit). The cytoplasmic and nuclear fractionation was performed using Ner-Per nuclear and cytoplasmic Kit (Thermo Fisher, Waltham, MA, USA) following the manufacturer’s instructions. The information about primary antibodies, including antibody dilution and source of antibodies in this study, is listed in Appendix A. The anti-Gli3 antibody can recognize both activate and repressor forms [28]. RhoGDI and PARP were used as an endogenous control for the cytoplasmic and nuclear fractions, respectively. The antigen–antibody complex was detected with Trident Femto western HRP substrate (GeneTex, Irvine, CA, USA). Specific protein bands were visualized using ChemiDoc XRS þ molecular imager (Bio-Rad, Hercules, CA, USA).

### 2.6. Migration Assay

LMS cells were seeded at 5 × 10^5^ per well in a 24-well plate and cultured overnight. When the cell density reached 100%, a straight scratch was created with a 200 µL pipette tip held perpendicular to the bottom of the 24-well plate. The cells were then washed three times with PBS and cultured in serum-free medium to avoid the proliferation of the LMS cells with varied treatments. Results were expressed as the space between the edges of individual wounds every 24 h for 72 h in comparison with initial (start) time using Image J Software 1.8.0_172. This assay was performed for three timepoints in triplicate [29].

### 2.7. Apoptosis Assay

Apoptosis was determined by Annexin V staining (APC Annexin V Apoptosis Detection Kit with 7-AAD, BioLegend, San Diego, CA, USA) following the manufacturer’s instructions. A total of 8 × 10^3^ LMS cells were grown in 6-well plates. The cells were treated with SMO, GLI, and DNMT inhibitors for 72 h. Staurosporine was used as a positive control. The Annexin V stain was evaluated using flow cytometry (Beckman Coulter Gallios, Indianapolis, IN, USA).

### 2.8. SMO Gene Silencing

A total of 5 × 10^5^ LMS cells were seeded in six-well plates for transfection. Lipofectamine RNAiMAx reagent (Invitrogen, Carlsbad, CA, USA) was used to transfect small interfering RNAs of *SMO* (esiRNA SMO, Sigma Aldrich St. Louis, MO, USA). RNA and protein expression levels were determined at 72 h post-transfection following the manufacturer’s instruction.

### 2.9. PTCH1 Methylation

A total of 8 × 10^4^ LMS cells were seeded in six-well plates and treated with DNMT inhibitor. The methylation status of the *PCTH1* promoter region was determined using the Human cancer EpiTect Methyl II PCR Array^®^ (QIAGEN Sciences, Frederick, MD, USA) following the manufacturer’s instructions. The ABI 7500 system for real-time PCR was used to read the plates. The relative amount of methylated and unmethylated DNA was calculated using the standard ΔCt method using an Excel spreadsheet provided by the manufacturer.

### 2.10. Statistical Analysis

Comparison of two groups was carried out using Student’s *t*-test for parametric distribution and Mann–Whitney test for nonparametric distribution. Comparison of multiple groups was carried out by analysis of variance (ANOVA) followed by a post-test using Tukey for parametric distribution and Kruskal–Wallis test followed by a post-test Dunns for nonparametric distribution, using GraphPad Prism 5 Software. Data were presented as mean ± standard error (SE). The significant difference was defined as *p* < 0.05.

## 3. Results

### 3.1. Hedgehog is Activated in LMS with Increased GLI Nuclear Translocation

The constitutive (basal) expression levels of HH signaling components in UTSM, HuLM, and LMS cells were evaluated by qRT-PCR. Higher expression of *SMO,* and *GLI1* was detected in LMS compared to UTSM (*p* < 0.05) and HuLM. The expression of *PTCH1* was down-regulated in LMS. *GLI2* did not show a difference among the cells, while full-length *GLI3* was upregulated in HuLM (Figure 1A), and the short form of GLI3 was not detected in all three cell lines. The protein expression levels of SMO and GLI1 were highest in LMS among three detected cell lines. The protein levels of GLI2 and GLI 3 were also upregulated in HuLM. The expression of HH ligands (*IHH, DHH,* and *SHH*) was not detected in these cells.

We further evaluated the level of GLIs nuclear translocation in UTSM, HuLM and LMS cells (Figure 1C). The expression levels of GLI1 were highest in the nucleus of the LMS cells, low in UTSM, and undetected in HuLM cells. GLI2 and GLI3 were mainly expressed in the nucleus of all cell lines, with the highest expression levels in LMS. (Figure 1C). These results showed that the HH pathway was activated in LMS due to the higher expression of SMO and GLI1 with increased GLI nuclear translocation.

### 3.2. Inhibition of the Hedgehog Pathway Using SMO and GLI Inhibitors in LMS Cells

SMO inhibitors (LDE225 and GDC0449) and GLI inhibitors (Gant58 and Gant61) were selected to determine their effect on LMS cells. MTT assay was performed using three time point (24, 48, and 72 h) with varying drug concentrations. Treatment with SMO inhibitor (LDE225) for 72 h showed a dose-dependent inhibitory effect on cell proliferation (Appendix A). A total of 10 µM showed 50% of the inhibitory effect and then was used in the following experiments. LMS cells treated with varying concentrations of GDC0449 did not show any inhibitory effect on cell proliferation (Appendix A). GLI1 inhibitor (Gant61) showed a dose-dependent inhibitory effect on LMS after 72 h of treatment. Treatment with 30 µM presented a 50% inhibitory effect on LMS, and therefore was chosen for the next experiments. However, for Gant58 (GLI inhibitor), we did not detect an inhibitory effect on LMS proliferation. Based on MTT results, LDE225 (SMO inhibitor) and Gant61 (GLI inhibitor) were selected for the next experiments (Appendix A).

To verify the specificity of the effect of SMO (LDE225) and GLI inhibitors (Gant61) on LMS, we performed an MTT assay using the same doses and duration in both UTSM and HuLM cells (Appendix A). After 72 h of treatment, LDE225 and Gant61 showed inhibitory effects on UTSM cell proliferation. The SMO inhibitor did not show a significant growth inhibition in UTSM cells (Appendix A), while the GLI inhibitor showed a significant effect (*p <* 0.05) in UTSM (Appendix A); however, the proliferation rate was decreased more in LMS cells when compared with UTSM cells in response to the GLI inhibitor. SMO or GLI inhibitors did not show any inhibitory effect on HuLM cell proliferation (Appendix A). To better compare the inhibitory effect of HH inhibitors in three cell lines from normal tissues as well as benign and malignant uterine tumors, we generated graphs (Appendix A) from Appendix A highlighting that the HH inhibitors exhibited a dominantly inhibitory effect on LMS. There is a significant difference between LMS and HuLM treated with LDE225 (*p* < 0.05).

LMS cells were treated with an SMO inhibitor, LDE225, for 24, 48, and 72 h, with drug replacement every 24 h. RNA and protein expression of HH components were evaluated for all time points. qRT-PCR showed that the expression levels of *SMO, GLI1, GLI2,* and *GLI3* were significantly downregulated after 72 h LDE225 treatment (*p* < 0.05), while alteration of expression levels after 24 and 48 h treatment was not observed (Figure 2A). However, the protein levels of these key HH members were decreased in a time-dependent manner in response to the treatment of SMO inhibitor (LDE225), suggesting that LDE225 treatment altered the HH components more dominantly at the protein levels (Figure 2B). For the GLI inhibitor, qRT-PCR analysis demonstrated that RNA expression of *SMO, GLI1, GLI2,* and *GLI3* was not altered after Gant61 treatment (Figure 2C). However, protein expression of GLI1, GLI2, and GLI3 were decreased in response to Gant61 treatment (Figure 2D).

The GLIs nuclear translocation was evaluated in LMS cells treated with LDE225 or Gant61 inhibitor for 24, 48, 72 h, respectively. The results showed that translocation of GLIs into the nucleus was markedly decreased in three timepoint treatments with an SMO or GLI inhibitor as compared to the untreated control (Figure 2E,F). 

To determine the effects of treatment with LDE225 or Gant61 inhibitor on the proliferation of LMS cells, an MTT assay was performed to determine the effect of the treatment on cell proliferation. Our data demonstrated that the proliferation was decreased after treatment with SMO or GLI inhibitor (Figure 2G,H).

Cell migration is a multi-step process that plays an important role in tumorigenesis. To evaluate the effect of SMO and GLI inhibitors on the migration of LMS cells, a wound-healing assay was performed. The results showed that the SMO and GLI inhibitor significantly decreased the migration capacity in LMS cells (Figure 3A) (*p* < 0.05). An apoptosis assay was performed and demonstrated that the SMO and GLI inhibitors induced apoptosis in LMS cells (Figure 3B), and the GLI inhibitor showed a more potent effect compared to the SMO inhibitor.

To explore the possible synergistic or additive effect of combination treatment with SMO and GLI inhibitors, we evaluated their inhibitory effect on LMS cells using MTT assay. The results showed no synergism or additive effect using a combination of the treatments (Appendix A)

Treatment with SMO or GLI inhibitors individually showed an inhibitory effect on proliferation and migration while inducing apoptosis in LMS cells. The inhibition of GLI nuclear translocation was more potent using the SMO inhibitor. Thus, the SMO gene was selected for the knockdown. The knockdown of the SMO gene was performed using interference RNA and the HH components were evaluated in LMS cells. SMO protein expression was evaluated to verify the efficiency of the knockdown and was highly decreased after the knockdown (Figure 3C). Figure 3D shows that knockdown of SMO decreased the expression of SMO, GLI1, GLI2, and GLI3 (*p* < 0.05).

### 3.3. Inhibition of DNA Methyltransferase Regulated HH Signaling in LMS Cells

To evaluate the activation of the HH signaling pathway in LMS in the context of methylation regulation, we determined the expression of DNA methyltransferases (DNMT) in LMS and UTSM cells. Our studies showed that RNA expression of *DNMT1, DNMT3a,* and *DNMT3b* was upregulated in LMS compared to UTSM cells (*p* < 0.05) (Figure 4A). Accordingly, the protein expression of DNMT1 and DNMT3a were also increased in the LMS compared to UTSM cells (Figure 4B). Next, we determined whether inhibition of DNA methyltransferases with 5′-Aza-2′-Deoxycytidine (5′-Aza-dc) affects LMS cells. We performed MTT assay using a different dose at three time points (24, 48, and 72 h). After 72 h of treatment, 5′-Aza-dc at the concentration of 2 µM showed 50% inhibition in proliferation (Appendix A). *PTCH1* DNA methylation was evaluated to verify if the treatment with DNA methyltransferase inhibitor was able to reverse the methylation profile in LMS cells. The basal level of the percentage of *PTCH1* DNA methylation in LMS was 2.3%. The percentage of *PTCH1* DNA methylation after 72 h of 5′-Aza-dc treatment was decreased to 1%.

RNA and protein expression levels of DNMTs were evaluated in LMS cells in response to 5′-Aza-dc treatment. The RNA expression of *DNMT3a* and *DNMT3b* was significantly decreased in 5′-Aza-dc-treated LMS cells compared to the control (*p* < 0.05) with 48 and 72 h of the treatment (Figure 4C). The protein expression of DNMT1 and DNMT3a in LMS cells was decreased after 72 h of the treatment with 5′-Aza-dc (Figure 4D).

To evaluate the impact of DNA methyltransferase inhibition on the HH signaling pathway, the RNA and protein levels of HH signaling components were evaluated in the presence or absence of 5′-Aza-dc in LMS cells. Although the RNA expression of *PTCH1, SMO, GLI2,* and *GLI3* was not altered after 5′-Aza-dc treatment, the decreased RNA expression of *GLI1* was observed in response to 5′-Aza-dc treatment (Figure 5A). WB analysis exhibited decreased expression levels of SMO and GLI1 (Figure 5B). Moreover, GLI1 and GLI2 nuclear translocation were decreased in response to 5′-Aza-dc treatment. On the other hand, 5′-Aza-dc treatment increased the nuclear translocation of GLI3 (Figure 5C).

We then evaluated the effect of DNMT inhibition on proliferation, migration, and apoptosis in LMS cells. The results showed that 5′-Aza-dc decreased proliferation (Figure 5D), concomitantly with decreased expression of PCNA (proliferation marker) in LMS cells (Appendix A). Migration capacity was decreased after 5′-Aza-dc treatment (*p* < 0.05) (Figure 5E). Moreover, 5′-Aza-dc treatment was capable of inducing apoptosis in LMS cells (Figure 5F).

### 3.4. Inhibition of Both DNA Methylation and Hedgehog Signaling in Human LMS Cell Lines

The treatment with DNA methyltransferase and HH inhibitors showed an inhibitory effect on HH signaling via decreasing GLI1 transcription and protein expression, as evidenced by decreasing proliferation and migration, while inducing apoptosis. Next, we performed experiments to explore whether the DNA methyltransferase inhibitor, in combination with HH inhibitors, could exhibit an additive or synergistic effect in LMS cells.

MTT assay was performed to evaluate the combined effect of DNA methylation and HH inhibitors on LMS proliferation. The results showed that the combination treatment with DNA methylation and SMO inhibitors did not show synergism or an additive effect. However, the combination of 5′-Aza-dc with GLI inhibitor showed a synergistic effect (Appendix A). Since the treatment of 1 µM 5′-Aza-dc with 30 µM GLI inhibitor exhibited a most potent inhibitory effect on LMS proliferation, this combination treatment was then used for further studies (Appendix A).

The RNA expression of key HH members, including *SMO, GLI1, GLI2,* and *GLI3* was measured with and without combination treatment. Our data demonstrated that combination treatment decreased the RNA expression of *SMO, GLI1, GLI2,* and *GLI3* (*p* < 0.05) compared to the control (Figure 6A). The combination treatment also resulted in decreased protein levels of *GLI1* (Figure 6B). Moreover, the combination treatment decreased GLI1 nuclear translocation in a time-dependent manner compared to the control (Figure 6C).

To explore the effect of GLI and 5′-Aza-dc inhibitors on LMS cells, the cells were treated with both inhibitors for 72 h, and proliferation was evaluated every 24 h. Figure 6D showed that combination treatment decreased the proliferation of LMS cells. In addition, the wound-healing assay demonstrated that combination treatment decreased the migration capacity in the LMS cells (*p* < 0.05) (Figure 6E). The combination showed a more potent effect than the single treatment (Appendix A), with decreased expression of HH signaling components, proliferation, and decreasing GLI1 nuclear translocation and migration capacity in LMS.

## 4. Discussion

Uterine leiomyosarcoma is a rare but extremely aggressive tumor that represents a treatment challenge due to its unresponsiveness to available therapies. As a consequence, the patients commonly present high rates of tumor recurrence, progression, and metastasis [5]. Previously, it was demonstrated that the protein levels of SMO and GLI1, the key members of HH signaling, were highly expressed in human uterine LMS [6]. However, how HH signaling is activated and its contribution to LMS malignant is still unknown. In this study, we used three cell lines representing the spectrum from a normal cell line (UTSM), benign uterine tumor cell line (HuLM), and uterine malignant cell line (LMS) to better understand the tumor progression linking to the HH signaling pathway. We demonstrated, for the first time, that HH is dominantly activated in LMS cells via the upregulation of key HH members concomitantly with increased nuclear translocation of GLI1. Importantly, DNA methylation is involved in the HH pathway activity in LMS cells. Moreover, targeting DNA methylation and HH pathways exhibited a potent inhibitory effect on LMS cells.

It has been reported that cilium, as an orphan organelle, was involved in the regulation of cellular events, including cell cycle and proliferation linking to HH pathway [30,31]. An electron microscopic study showed that single cilia with centriolar apparatus were encountered in HuLM, but were absent in LMS cells [31]. Notably, in our study, the constitutive upregulation of HH components was more dominant in LMS cells. In this regard, the impact of the absence of cilium on constitutive activation of HH in LMS needs to be further investigated.

SMO, GLI, and PTCH1 for the HH key members were selected for evaluation as triggers of the HH signaling in LMS cells. LDE225 and Gant61 were used in this study because they are already established as the potent inhibitors of SMO and GLI, respectively. Although both drugs had suppressed the LMS cells proliferation, LDE225 showed a more potent effect in decreasing GLI nuclear expression than Gant61 treatment.

The anti-tumor effect of LDE225 observed in our model was consistent with the literature for other types of cancer. For instance, in renal carcinoma cells, treatment with LDE225 showed reduced cell proliferation, concomitant with lower GLI1 and GLI2 expression [32]. In melanoma cells, treatment with LDE225 showed a decrease in apoptosis rate and cell proliferation, as well as an increase in cell cycle arrest. In the melanoma animal model, treatment with LDE225 inhibited GLI1 expression [33]. In Hepatoma cells, the treatment suppressed cell proliferation and decreased the protein levels of the GLI2 and ABCC1 transporter [34]. In lung cancer, the combined treatment with LDE225 and Erlotinib (an EGFR inhibitor) showed a reduction in cell invasion, migration, colony formation, proliferation, and induced apoptosis [35]. In chronic myeloid leukemia, LDE225 treatment inhibited cell growth with GLI1 downregulation [36]. Studies in LMS and other types of cancers using LDE225 demonstrated that targeting SMO is sufficient to suppress the pathogenesis of these aggressive tumors. In addition, we observed that the decrease in RNA expression of HH components occurred after 72 h treatment with an HH inhibitor (LDE225). On the other hand, the decreased protein levels of HH components such as SMO and GLI1 were observed after treatment with LDE225 for only 24 h, suggesting that LDE225 treatment altered the HH components more dominantly at the protein levels. This is consistent with the previous report that LDE225 interacts with SMO in the drug-binding pocket, where it acts as an antagonist, preventing downstream activation of HH signaling [37]. In addition, as shown in Appendix A, LDE225 treated LMS did not show a cell morphology change as compared to the untreated cells at three time points (24, 48, and 72 h), suggesting that LDE225 treatment does not alter the LMS phenotype via cell toxicity.

Targeting GLI1 with Gant61 has been studied in other types of cancers, showing beneficial effects. In breast cancer, Gant61 treatment decreased cell proliferation by reducing GLI1 and PTCH1 gene expression and inhibited GLI1 nuclear translocation [38]. In vivo studies showed that Gant61 treatment reduced both tumor growth and GLI1 expression in rhabdomyosarcoma [39]. In prostate cancer, treatment with Gant61 induced suppression of tumor growth with decreased GLI1 and PTCH1 expression [40]. Our results showed that Gant61 is able to impair GLI protein expression, reduce GLI1 nuclear translocation, and exhibit an inhibitory effect on LMS cells’ proliferation. All those studies demonstrated the important role of GLIs in cancer development and targeting GLIs in several types of neoplasms.

LMS has high recurrence and metastasis rates [41]. Therefore, evaluation of the effects of SMO and GLI inhibitors in the migration process was extremely important since this mechanism is directly involved with the metastasis and progression of the disease. Regarding this, our results showed that LDE225 and Gant61 have an inhibitory activity in cell migration. This inhibitory effect by HH inhibitors was consistent with studies in other types of tumors [32,42,43,44]. In prostate cancer, Gant61 and LDE225 treatment showed a decrease in the migration capacity of the cells [45,46]. Gant61 treatment in ovarian cancer cells showed a reduction in migration with downregulation of GLI1 activity [47]. In glioblastoma cells, treatment with Gant61 suppressed migration with impairment of the expression of HH components [48]. Meanwhile, we observed that LDE225 and Gant61 inhibited LMS proliferation and induced apoptosis, which indicated that decreased migration may be partially due to the decreased number of the cells in response to LDE225 and Gant61 treatment. Indeed, consistent with our finding, Gant61 and LDE225 have been reported to induce apoptosis in other types of cancers. In pancreatic cancer, Gant61 induces apoptosis with downregulation of GLI expression [49]. In prostate cancer, Gant61 induced apoptosis [50]. In melanoma cells, LDE225 increased the percentage of apoptotic cells, inhibited cell proliferation, and reduced the expression of HH pathway components [51].

Inhibition of SMO or GLI using LDE225 and Gant61, respectively, blocked the HH pathway in LMS, decreasing proliferation, migration, and inducing apoptosis. In addition, our results showed that in the basal level of gene expression, the downregulation of *PTCH1* in LMS might indicate a possible aberrant CpG island hypermethylation of the *PTCH1* gene. Activation of the HH signaling pathway can also occur due to methylation. The PTCH1 promoter region has been shown to be hypermethylated in rhabdomyosarcoma, medulloblastoma [20], and breast cancer [21,22]. The treatment with 5′-Aza-dc in rhabdomyosarcoma and medulloblastoma showed a decrease in *PTCH1* gene methylation [20]. However, we did not find changes in *PTCH1* expression after 5′-Aza-dc treatment.

Interestingly, the treatment with 5′-Aza-dc in LMS regulated GLI1 expression, decrease proliferation, migration, and inducing apoptosis. The mechanism by which 5′-Aza-dc regulates GLI1 expression in LMS is not understood. It may occur due to the indirect regulation of GLI1 expression by crosstalk with another signaling (AKT1, TGFβ, WIP1 and HDAC), therefore activating GLI1 expression.

We explored the effect of combination treatment using 5′-Aza-dc and Gant61. The results showed a synergistic effect with those drugs. Combination treatment exhibited a more potent effect compared to single treatment in the context of inhibiting the HH pathway and affecting LMS cells’ proliferative and migratory behavior. The combination treatment with HH and epigenetic inhibitors has been shown to exhibit a more potent effect compared to a single treatment in other types of tumors. In liver cancer, SMO and HDAC inhibitors showed decreased cell viability, colony formation, and increased apoptosis [52]. In aerodigestive cancer, the combined treatment with SMO and HDAC inhibitors promotes cell cycle arrest, suppresses SMO and PTCH1 expression, and delays tumor growth in an animal model, prolonging survival more than a single agent alone [53]. Our study, using a combination treatment strategy to produce synergistic activity, is one possible advantage in achieving higher cure rates in LMS.

We proposed a mechanism model of HH pathway activation in LMS based on our novel findings that (1) SMO and GLI expression are deregulated in LMS cells, (2) GLI nuclear translocation is increased in LMS cells, (3) epigenetic deregulation is involved in GLI1 expression, and (4) targeting the HH pathway via pharmacological inhibition is able to suppress the LMS phenotype (Figure 6F).

## 5. Conclusions

Our studies demonstrated for the first time that HH signaling is activated in LMS. We also demonstrated that pharmacological suppression of SMO and GLI was capable of inhibiting LMS cell proliferation, migration, and invasion. Besides, the DNMTi treatment alone or in combination with the GLI inhibitor had a more potent effect on the LMS cells. All these data open new perspectives for uterine management, focusing on the development of novel, non-invasive specific therapeutics for this aggressive tumor.

## Figures and Tables

**Figure 1 cells-10-00053-f001:**
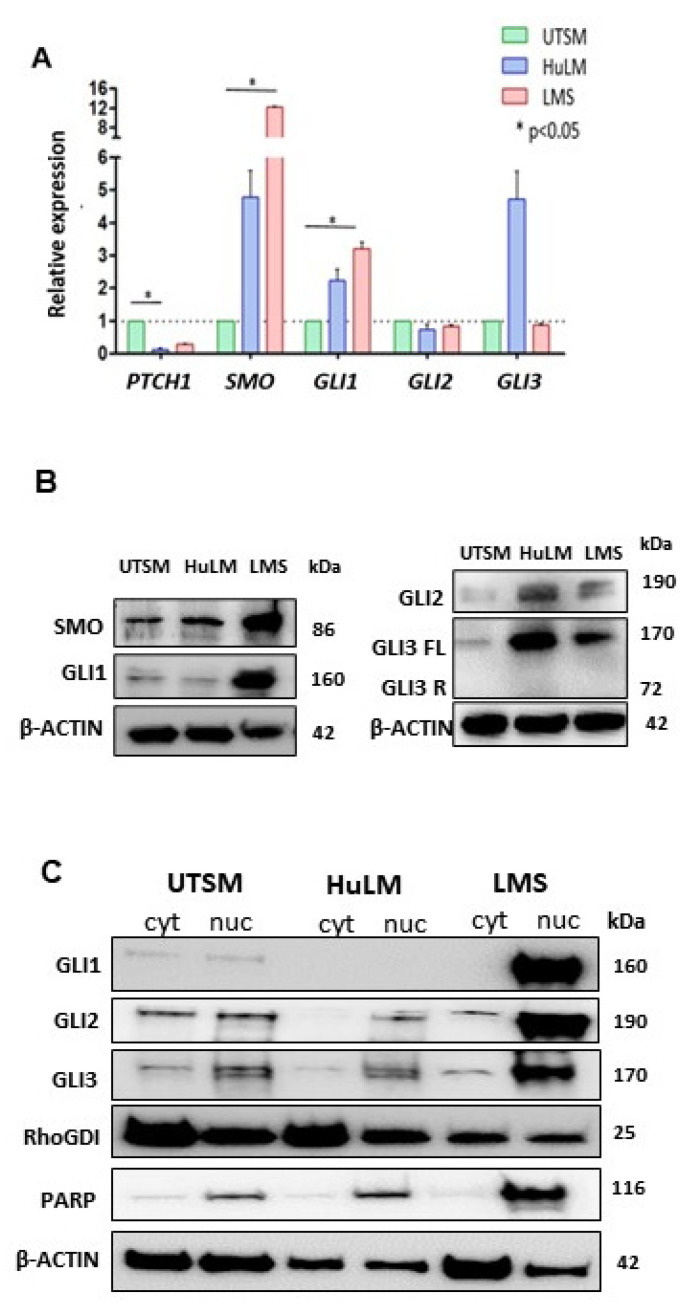
Basal Hedgehog (HH) signaling components gene and protein expression in uterine smooth muscle (UTSM), human leiomyoma cell line (HuLM), and uterine leiomyosarcoma (LMS) cell lines. (**A**) mRNA expression levels of *PTCH1, SMO, GLI1, GLI2*, and *GLI3* quantified by qRT-PCR. Relative expression values were obtained after reference and endogenous control normalization. (**B**) Protein expression of SMO, GLI1, GLI2, and GLI3FL and GLI3R. (**C**) Protein expression of GLI1, 2, and 3 in both cytoplasm and nucleus compartment of the cells. * *p* < 0.05.

**Figure 2 cells-10-00053-f002:**
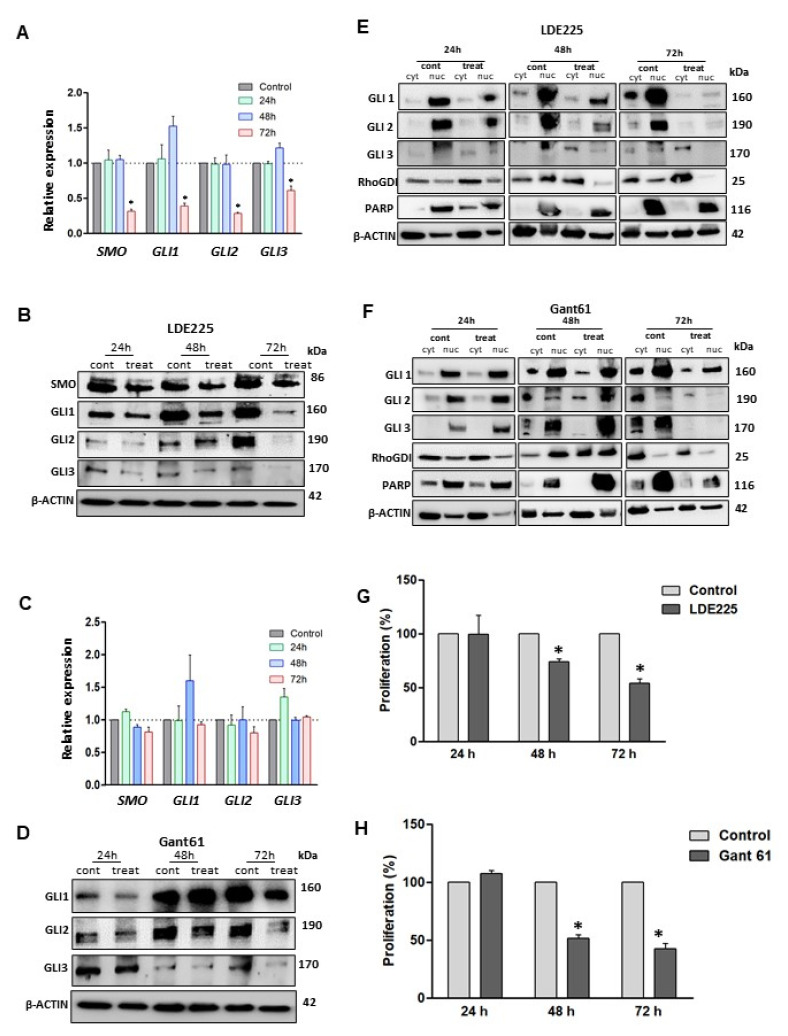
Treatment with HH inhibitors decreased transcript and protein levels and reduced LMS cell proliferation. (**A**) RNA expression of HH components (*SMO, GLI1, GLI2,* and *GLI3*) using LDE225, (**B**) Western blot images showing protein expression profile of SMO, GLI1, GLI2 and GLI3 after SMO inhibition (LDE225). (**C**) RNA expression of HH components after GLI inhibition. (**D**) Protein expression of GLI1, GLI2, and GLI3 after Gant61 treatment. (**E**,**F**) GLIs proteins expression in the cytoplasm and nucleus compartments after 72 h of treatment with LDE225 and Gant61, respectively. (**G**,**H**) MTT assays comparing the proliferation profile of LMS cells after treatments at different time points (24, 48, and 72 h). * *p* < 0.05.

**Figure 3 cells-10-00053-f003:**
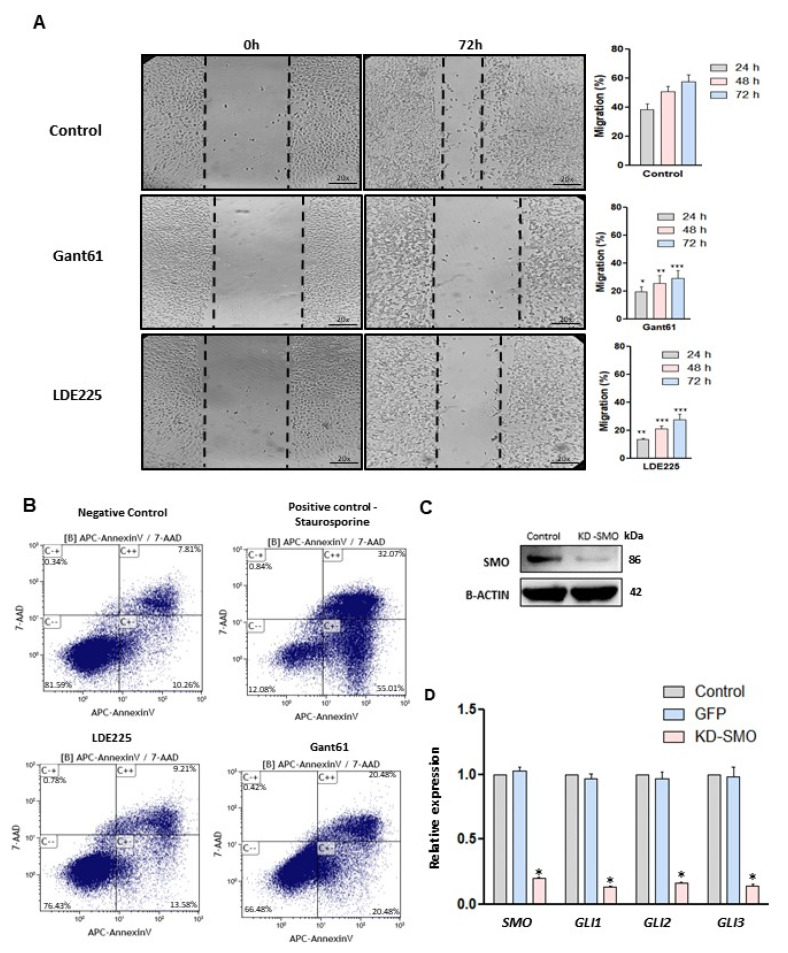
Effect of HH inhibitors on migration and apoptosis in LMS cells and knockdown of SMO. (**A**) Wound healing assay (20×), the bar chart showing quantitative analysis of cell migration after 72 h of treatment compared to the control (no treated). (**B**) Apoptosis rates of LMS cell (annexin V assay) after 72 h with SMO or GLI inhibitors. Staurosporine treatment was used as a positive control (24 h). (**C**) SMO protein expression after *SMO* gene knockdown. (**D**) RNA expression of HH components after *SMO* gene knockdown. * *p* < 0.05, ** *p* < 0.01, *** *p* < 0.001.

**Figure 4 cells-10-00053-f004:**
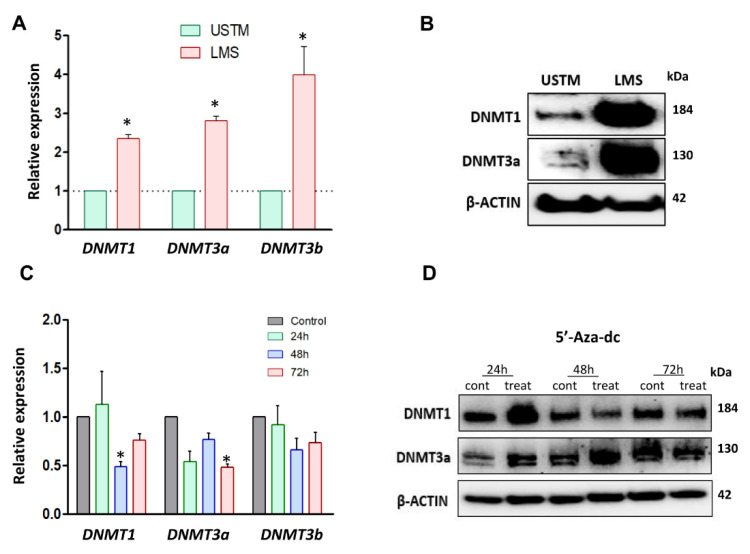
DNMTs expression levels in LMS cells, and the effects of DNMT inhibitor. (**A**) mRNA levels of *DNMTs* (*DNMT1, DNMT3a, and DNMT3b*) in UTSM and LMS. (**B**) Protein expression of DNMT1 and DNMT3a in UTSM and LMS by Western blot. (**C**) RNA expression of *DNMT1, DNMT3a, and DNMT3b* in response to 5′-Aza-dc treatment for 72 h. (**D**) Protein expression of DNMTs (DNMT1 and DNMT3a) in response to 5-Aza-dc treatment for 72 h, and quantification analysis of DNMTs protein. * *p* < 0.05.

**Figure 5 cells-10-00053-f005:**
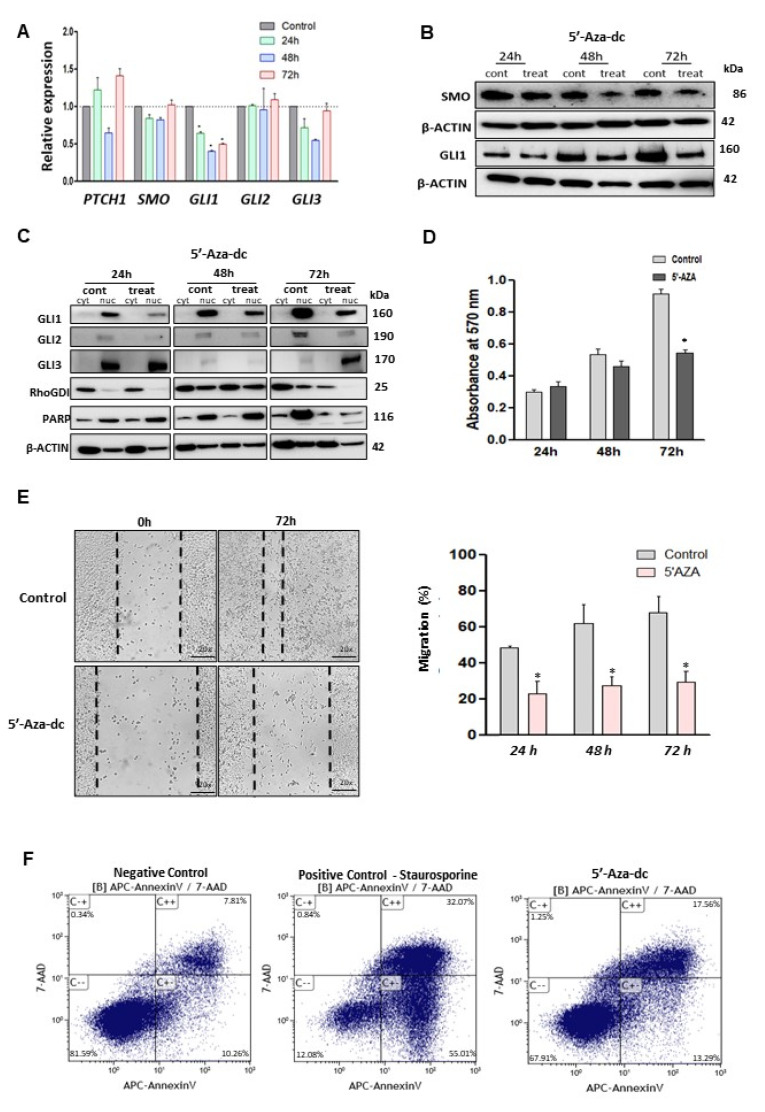
DNMTs inhibition and its effects on GLI1 expression, cell proliferation, cell migration and apoptosis induction. (**A**) *PTCH1, SMO, GLI1, GLI2*, and *GLI3* expression after 5′-Aza-dc cell treatment compared to control (no treated cells). (**B**) Protein expression of SMO and GLI1 with or without 5′-Aza-dc treatment. (**C**) GLIs protein expression in cytoplasm and nuclear cell compartment, after 24, 48, and 72 h of 5′-Aza-dc treatment. (**D**) MTT assay in the presence or absence of 5′-Aza-dc for 72 hs. (**E**) Wound healing assay (20×) for cell migration measured after 72 h of 5′Aza treatment, and quantification analysis of percentage of migration. (**F**) Apoptosis assay using annexin V stain after 72 h of 5′-Aza-dc treatment. Staurosporine treatment for 24 h was used as a positive control. * *p* < 0.05.

**Figure 6 cells-10-00053-f006:**
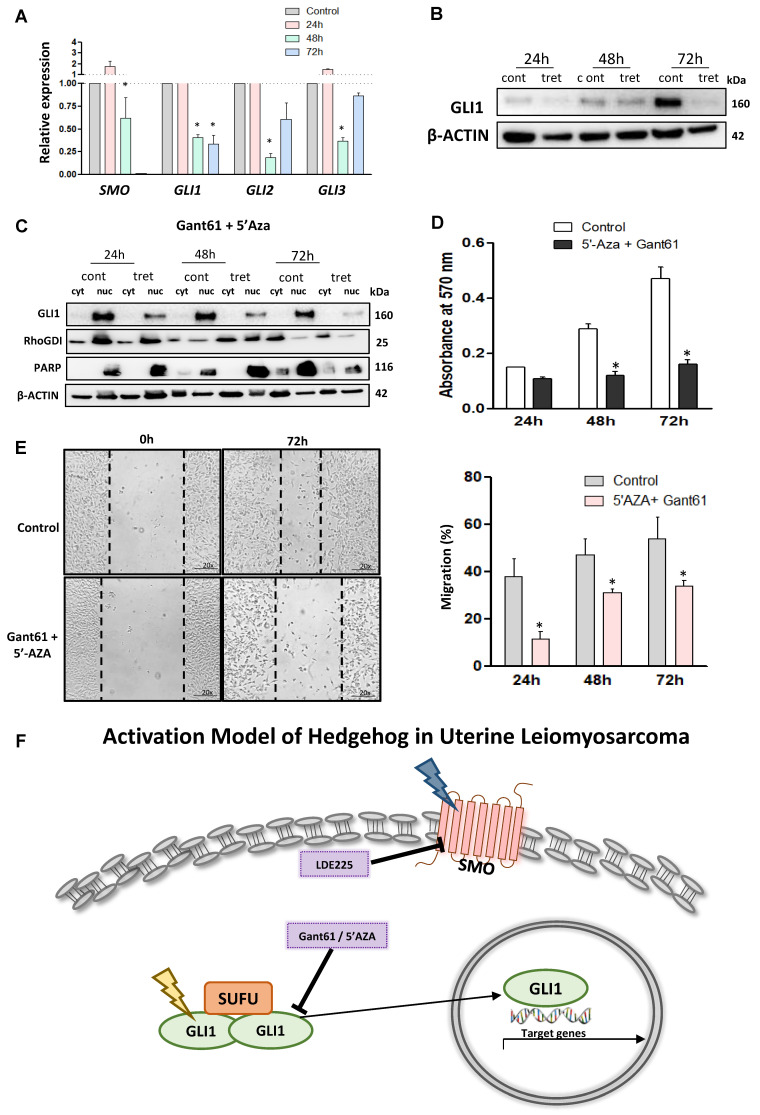
Effects of combination treatment with 5′-Aza-dc and Gant61 on HH signaling in LMS cells. (**A**) RNA expression of *SMO, GLI1, GLI2,* and *GLI3* with combination treatment or without treatment. (**B**) GLI1 protein expression in the presence or absence of the combination treatment. (**C**) GLI1 protein expression in the cytoplasm and nuclear compartment. (**D**) Measurement of cell proliferation using the MTT assay. (**E**) Wound healing assay after 72 h of the combination treatment and quantification analysis of the percentage of migration. * *p* < 0.05. (**F**) Proposed model of HH signaling pathway activation in uterine LMS cells. The expression of SMO and GLI is deregulated and GLI nuclear translocation is increased, and by targeting these molecules we were able to inhibit the LMS phenotype. We also proposed that the epigenetic mechanism is involved in the hyper-regulation of GLI1 expression. The deregulation of SMO and GLI1 is represented in yellow and blue, respectively.

## Data Availability

All data generated or analyzed during this study are included in this article, the Supplementary file and the published paper doi:10.21203/rs.3.rs-27677/v1.

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
