# Peer review of "Targeting Hedgehog Pathway and DNA Methyltransferases in Uterine Leiomyosarcoma Cells"

_cells, 2020, doi:10.3390/cells10010053_

Round 1
Reviewer 1 Report
Point 1: The authors present the results as if up/downregulation of any of the Hh actors they choose to study were equivalent and they never discuss the data in detail neither in the result nor in the discussion section. For example, in Figure 1 A (but this holds true throughout the article), Smoothened, is an activator of the downstream elements of the pathway, when SUFU is a repressor. Gli1 is a transcription factor which is an activator of the target genes, but only when the pathway is activated, when Gli2 and Gli3 are not upregulated by the activation of the pathway. Gli2 is the major activator transcription factor, when Gli3 is the major repressor in it’s truncated form…. For instance, can the authors explain why an activator like Smo is upregulated, but also a repressor like SUFU?
Response 1: Considering the basal levels of gene expression, As shown in Figure 1, the relative expression levels of SUFU transcripts in LMS cells were ~1 and ~2 times increases in HuLM compared to HuLM and UTSM, respectively. On the other hand, the expression of SMO showed 12 and 4.5 times higher in LMS than in UTSM and HuLM, respectively. At the protein levels, SMO shows an increasing expression profile from benign cells to malignant, and SUFU presents similar levels cross the cell lines. It is known that several mechanisms are involved in gene expression regulation [1,2]. Although we did not characterize the regulatory mechanisms of SMO and SUFU, we believe that cells might bear several compensatory mechanisms to ensure survival. In addition, the levels of SMO decreased after target therapy (Figure 2), which is more clinically relevant and consistent with our previous finding in human samples (3)
[1] George Orphanides, Danny Reinberg, A Unified Theory of Gene Expression, Cell, Volume 108, Issue 4, 2002, Pages 439-451, ISSN 0092-8674, doi.org/10.1016/S0092-8674(02)00655-4.
[2] Gorojankina T. Hedgehog signaling pathway: a novel model and molecular mechanisms of signal transduction. Cell Mol Life Sci. 2016 Apr;73(7):1317-32. doi: 10.1007/s00018-015-2127-4. Epub 2016 Jan 13. PMID: 26762301
[3] Garcia N, Bozzini N, Baiocchi G, da Cunha IW, Maciel GA, Soares Junior JM, et al. May Sonic Hedgehog proteins be markers for malignancy in uterine smooth muscle tumors? Hum Pathol. 2016;50:43-50. 2
SuFu was just an example, I could have commented on the fact that Gli3 expression is extremely elevated in HuLM cells and not in the other two cell lines. My point was that you present data throughout the paper that you do not comment or try to synthesize to make sense of it. In the specific case of SuFu, if you think SuFu is irrelevant, say it or take it off the graph, in my opinion, it complexifies the message that you are trying to get across. Especially since you never show it again in the rest of the manuscript.
Point 2: In the same line of thoughts, the authors study Gli3 expression by qPCR, which is not sufficient to decipher between its weak role as an activator and its far more potent role as a repressor since the difference is due to its proteolysis. Then, they perfom WBs which is a good way to check both the full length form (activator) and shorter form (repressor) and still, they fail at showing the repressor from (I’m guessing based on the molecular weight that what they show is the activator form, but it is not even stated…). Although I would agree that the pathway seems constitutively activated given the data for Smo and Gli1, but it’s hard to draw a definitive conclusion when the repressor part is occulted.
Response2: Thanks for the comment. The GLI3 full-length and shorter forms were evaluated in the basal level using the Gli3 antibody which can recognize both full-length and short forms of GLI3(Figure 1B, Original file), however the shorter form was not detected in all cell lines (UTSM, HuLM and LMS) we tested. We have included the new info in the manuscript (Page 10, lines 188-190).
If this is true this is extremely valuable information. Please modify figures and figure legends accordingly.
However, I have to underline that this is really bizarre that you can never detect Gli3R, especially in the UTSM cells! How do you make sense of this if your hypothesis is that aggressiveness of LMS cells are linked to constitutive activation of HH???
Moreover, if you look at your own data carefully, it’s clear that Gli3 mRNA expression in UTSM cells is comparable mRNA expression in LMS(Fig 1A) and yet, it displays way less Gli3FL (Fig 1B)… In my opinion, you should be trying another antibody, or at least verify that this one is indeed able to recognize the truncated Gli3R form in other , better characterized, cell lines…
Point 3: The authors show that the pathway is activated in these cells, but are exogenous activators able to crank up the activation or is the pathway already at full speed? There is no experiment showing what Hh activators (Shh, SAG, purmorphamine) do on these cells.
Response 3: Thanks for the comments. We previously reported that patients with LMS expressed lower levels of SHH ligand, the activator of HH pathway. The same result was observed in the cell lines we studied. Based on the fact that HH pathway is constitutively activated in LMS, we decided to inactivate the pathway. Here, our goal was to assess components expression profile of this pathway and determine the effects of target therapy in vitro. Our perspective is to perform studies on the effect of HH activators on LMS derived tumor formation and therapy in animal models to evaluated the features and mechanisms involved in HH regulation in induced ex-vivo model. Since this manuscript focused on constitutively active HH pathway in LMS, we will work on the effect of HH activators on the LMS in the future and will submit a manuscript separately.
I think the experiment would have been valuable if, as I said, your goal was to describe Hh in the detail in these cells. Since you change the way you state your goal in this revised version, this is now beyond the scope of the paper.
Point 4: I find it quite puzzling that in 2020, nearly 20 years after the discovery of the regulation of the Hedgehog pathway by the primary cilium, researcher would consider publishing a paper without even mentioning the cilium. Just a few questions that I feel should be answered if the aim of the study is really to understand the regulation of Hh in these cells : Are the cells even ciliated? If yes, is Smo in the cilium at the basal state? What about patched does it leave the cilium upon actiavtion? Are Glis properly localized in the cilium?
Response 4: Thanks very much for the comment. Although the cilium has not been found in the LMS, we have included a paragraph discussing the potential role of cilium linking to HH pathway in LMS
(see page 17, lines 338-343).
LMS Cells being unciliated, your comment in the discussion section is appropriate
Point 5: Since the characterization of the pathway in its basal state is weak it is very hard to draw conclusions when modulators are added, let alone try to make sense of DNA methylation profiles.
Response 5: Thanks for the comments. We would like to clarify that our study does not intend to establish the whole function or mechanisms involved in the HH pathway regulation. We know that several molecules and pathways interact to give rise to a tumor, but our studies show the first time that targeting HH pathway and DNA methylation suppresses the LMS phenotype. Further characterization of the mechanism underlying HH pathway related to LMS pathogenesis is needed. However, considering that LMS is a poorly understood tumor that presents higher rates of morbidity and mortality without effective option of treatment, our results might provide a better understanding of this malignancy and help to development of new treatment strategies.
OK
Point 6: The authors don’t explain the relevance of studying the three cell lines. As clinicians, I think their expertise in this specific cancer is highly valuable, so please explain why you are comparing HuLM to LMS cells. What were you expecting using this cell line in addition to cells from normal tissue? Also in a minor point, the acronyms for these cell lines are not in the main text, only in the Material and methods…
Response 6: Thanks for the comments. We have included the explanation why we were using three kinds of cell lines in the manuscript (Page 16-17, lines 330-333) and have the acronyms for these cell lines throughout the whole manuscript (Page 5, line 99-102).
Thank you, but I think it would help the reader to understand what the different cell types are used for before the result section as well. This way when one dives into the manuscripts one can readily interpret the result of your experiments.
Point 7: Some conclusions seem to be a little biased. For example, about the repressor drug used in fig S3 and S4 the text states that the decrease in proliferation is more potent in LMS celle compared to UTSM, when the dose-reponse graphs for LDE225 and GDC0449 look very similar for the two cell lines.
Response7: Thanks for the comments. To better compare the inhibitory effect of HH inhibitors in three cell lines we tested, first we regenerated the curve graphs instead of bar figures to better present the dose and time dependent inhibitory effect using HH inhibitors. Secondly we generated new Figures (Figure S4 E and F), which clearly showed that HH inhibitors exhibited a dominantly inhibitor effect on LMS. We have included the new Figure 4E, F as well as corresponding write-up in the manuscript (Page11, Lines 220-225).
This new representation is much clearer. Maybe you should try to find a way to remove the gap between the Y axis and the 24h timepoints, as the reader who is not careful might not have seen that what is displayed is a relative measurement (proliferation) and then wonder where is the T0 timepoint..
Point 8: Figure 2. I find it doubtful that all the Hh actors considered in the panels A and B would all decrease in the same range and at the latest time point (72h) when treated with LDE225. Are you sure your cells remain healthy with 72h of treatment with that drug? Could you perform a cell toxicity assay (LDH??) ?
Response 8: Thanks very much for the comments. Although the decrease in RNA expression of HH components occurred after 72 hs treatment with HH inhibitor (LDE225), decreased protein levels of HH components such as SMO and GLI1 were observed after treatment with LDE225 for only 24 hs, suggesting that LDE225 treatment altered the HH components more dominantly at protein levels. This is consistent with previous report that LDE225 interacts with SMO in the drug-binding pocket, where it acts as an antagonist, preventing downstream activation of HH signaling (Jain S et al, 2017, OncoTargets and Therapy). In addition, as shown in Figure S8, LDE225 treated LMS did not show cell morphology change as compared to the untreated cells at three time points ( 24 hs, 48 hs, and 72 hs), suggesting that LDE225 treatment alters the LMS phenotype not via cell toxicity. Per reviewer’s comment, we emphasized our statement and included it in the manuscript ( page 12, lines 232-233, page 18, lines 368-371).
OK
Point 9: I do not understand why the values for your controls are getting higher with time in panels G and H? It is not the same thing to have a control staying at the same level and see a decrease of 4
the treated condition and to see the controls getting higher without explanation when the treated conditions remain constant, especially if after 24h of treatment there is no visible difference.
Response 9: The graphics show data of cell proliferation analyses and they were created using the absorbance values at 570 nm from all the viable cells. It was expected that untreated cells (control group) showed increasing values over time because they kept growing.
OK, then for clarity’s sake, please normalize to control.
Point 10: To conclude, despite the great amount of work behind this study, I think the overall issue is that the scientific question the authors are trying to answer is unclear. If the goal was to assess the role of the Hh pathway in these cells, I would advise to review the data with a Hh specialist to try to make sense of what has been done so far and design better experiments to complete the study (Gli3R, Primary cilium, better interpretation of the data…)
Response 10: Thanks for the comments. In this manuscript, we focused on determining the effects of intervention strategies, including targeting HH pathway and DNA methylation on HH components expression and phenotype in LMS. Further studies will be performed to better clarify the role of this pathway in LMS using in vivo model.
Point 11: If the aim of the study is to get a feel of the Hh pathway as a therapeutic target in this specific cancer, I would advise to go deeper in the characterization of the phenotype associated with the diverse drugs by using more sensitive techniques to add to the data (KI67or BrDU instead of MTT, 3D migration instead of wound healing, Cell cycle analysis, maybe grafted animals…)
Response 11: Thanks for the comments. Although we have several important findings in the manuscript as illustrated in 6 figures and 8 supplemental figures, we agreed with reviewer’s comments that further studies are needed, such as animal work, new and alternative approaches etc., to better understand the role of HH pathway in LMS. We will design all related experiments and hopefully generate exciting data for future manuscripts.
I think the overall quality of the manuscript has increase substantially and the goal is clearer. The main point remaining to solve is what happens with your Gli3 antibody and Gli3R. It is a major issue with the paper, since you’re using Gli3 as a readout of the activation of the pathway and drug activity in almost all of your experiments…
Author Response
Point 1: The authors present the results as if up/downregulation of any of the Hh actors they choose to study were equivalent and they never discuss the data in detail neither in the result nor in the discussion section. For example, in Figure 1 A (but this holds true throughout the article), Smoothened, is an activator of the downstream elements of the pathway, when SUFU is a repressor. Gli1 is a transcription factor which is an activator of the target genes, but only when the pathway is activated, when Gli2 and Gli3 are not upregulated by the activation of the pathway. Gli2 is the major activator transcription factor, when Gli3 is the major repressor in it’s truncated form…. For instance, can the authors explain why an activator like Smo is upregulated, but also a repressor like SUFU?
Response 1: Considering the basal levels of gene expression, As shown in Figure 1, the relative expression levels of SUFU transcripts in LMS cells were ~1 and ~2 times increases in HuLM compared to HuLM and UTSM, respectively. On the other hand, the expression of SMO showed 12 and 4.5 times higher in LMS than in UTSM and HuLM, respectively. At the protein levels, SMO shows an increasing expression profile from benign cells to malignant, and SUFU presents similar levels cross the cell lines. It is known that several mechanisms are involved in gene expression regulation [1,2]. Although we did not characterize the regulatory mechanisms of SMO and SUFU, we believe that cells might bear several compensatory mechanisms to ensure survival. In addition, the levels of SMO decreased after target therapy (Figure 2), which is more clinically relevant and consistent with our previous finding in human samples (3)
[1] George Orphanides, Danny Reinberg, A Unified Theory of Gene Expression, Cell, Volume 108, Issue 4, 2002, Pages 439-451, ISSN 0092-8674, doi.org/10.1016/S0092-8674(02)00655-4.
[2] Gorojankina T. Hedgehog signaling pathway: a novel model and molecular mechanisms of signal transduction. Cell Mol Life Sci. 2016 Apr;73(7):1317-32. doi: 10.1007/s00018-015-2127-4. Epub 2016 Jan 13. PMID: 26762301
3] Garcia N, Bozzini N, Baiocchi G, da Cunha IW, Maciel GA, Soares Junior JM, et al. May Sonic Hedgehog proteins be markers for malignancy in uterine smooth muscle tumors? Hum Pathol. 2016;50:43-50. 2
SuFu was just an example, I could have commented on the fact that Gli3 expression is extremely elevated in HuLM cells and not in the other two cell lines. My point was that you present data throughout the paper that you do not comment or try to synthesize to make sense of it. In the specific case of SuFu, if you think SuFu is irrelevant, say it or take it off the graph, in my opinion, it complexifies the message that you are trying to get across. Especially since you never show it again in the rest of the manuscript.
Response Thanks for the comment. Per reviewer’s suggestion, SuFu was removed from the entire manuscript.
Point 2: In the same line of thoughts, the authors study Gli3 expression by qPCR, which is not sufficient to decipher between its weak role as an activator and its far more potent role as a repressor since the difference is due to its proteolysis. Then, they perfom WBs which is a good way to check both the full length form (activator) and shorter form (repressor) and still, they fail at showing the repressor from (I’m guessing based on the molecular weight that what they show is the activator form, but it is not even stated…). Although I would agree that the pathway seems constitutively activated given the data for Smo and Gli1, but it’s hard to draw a definitive conclusion when the repressor part is occulted.
Response 2: Thanks for the comment. The GLI3 full-length and shorter forms were evaluated in the basal level using the Gli3 antibody which can recognize both full-length and short forms of GLI3(Figure 1B, Original file), however the shorter form was not detected in all cell lines (UTSM, HuLM and LMS) we tested. We have included the new info in the manuscript (Page 10, lines 188-190).
If this is true this is extremely valuable information. Please modify figures and figure legends accordingly
Response The corresponding figure and figure legends were modified (Page 5, line 170).
However, I have to underline that this is really bizarre that you can never detect Gli3R, especially in the UTSM cells! How do you make sense of this if your hypothesis is that aggressiveness of LMS cells are linked to constitutive activation of HH???
Moreover, if you look at your own data carefully, it’s clear that Gli3 mRNA expression in UTSM cells is comparable mRNA expression in LMS(Fig 1A) and yet, it displays way less Gli3FL (Fig 1B)… In my opinion, you should be trying another antibody, or at least verify that this one is indeed able to recognize the truncated Gli3R form in other , better characterized, cell lines…
Response: Thanks for the comment. The GLI3 antibody we used in this manuscript can recognize both activate and repressor form, which has been confirmed in other type of cells. We have cited a paper using the same antibody which can detect both activate and repressor forms (Reference 28, see reference below as well)). In LMS cells, the only activate form is observed.
Matz-Soja M, Rennert C, Schönefeld K, Aleithe S, Boettger J, Schmidt-Heck W, et al. Hedgehog signaling is a potent regulator of liver lipid metabolism and reveals a GLI-code associated with steatosis. Elife. 2016;5.
Point 3: The authors show that the pathway is activated in these cells, but are exogenous activators able to crank up the activation or is the pathway already at full speed? There is no experiment showing what Hh activators (Shh, SAG, purmorphamine) do on these cells.
Response 3: Thanks for the comments. We previously reported that patients with LMS expressed lower levels of SHH ligand, the activator of HH pathway. The same result was observed in the cell lines we studied. Based on the fact that HH pathway is constitutively activated in LMS, we decided to inactivate the pathway. Here, our goal was to assess components expression profile of this pathway and determine the effects of target therapy in vitro. Our perspective is to perform studies on the effect of HH activators on LMS derived tumor formation and therapy in animal models to evaluated the features and mechanisms involved in HH regulation in induced ex-vivo model. Since this manuscript focused on constitutively active HH pathway in LMS, we will work on the effect of HH activators on the LMS in the future and will submit a manuscript separately.
I think the experiment would have been valuable if, as I said, your goal was to describe Hh in the detail in these cells. Since you change the way you state your goal in this revised version, this is now beyond the scope of the paper.
Response: Thanks very much for the comments
Point 4: I find it quite puzzling that in 2020, nearly 20 years after the discovery of the regulation of the Hedgehog pathway by the primary cilium, researcher would consider publishing a paper without even mentioning the cilium. Just a few questions that I feel should be answered if the aim of the study is really to understand the regulation of Hh in these cells : Are the cells even ciliated? If yes, is Smo in the cilium at the basal state? What about patched does it leave the cilium upon actiavtion? Are Glis properly localized in the cilium?
Response 4: Thanks very much for the comment. Although the cilium has not been found in the LMS, we have included a paragraph discussing the potential role of cilium linking to HH pathway in LMS
(see page 17, lines 338-343).
LMS Cells being unciliated, your comment in the discussion section is appropriate
Response: Thanks very much for the comments
Point 5: Since the characterization of the pathway in its basal state is weak it is very hard to draw conclusions when modulators are added, let alone try to make sense of DNA methylation profiles.
Response 5: Thanks for the comments. We would like to clarify that our study does not intend to establish the whole function or mechanisms involved in the HH pathway regulation. We know that several molecules and pathways interact to give rise to a tumor, but our studies show the first time that targeting HH pathway and DNA methylation suppresses the LMS phenotype. Further characterization of the mechanism underlying HH pathway related to LMS pathogenesis is needed. However, considering that LMS is a poorly understood tumor that presents higher rates of morbidity and mortality without effective option of treatment, our results might provide a better understanding of this malignancy and help to development of new treatment strategies.
OK
Response: Thanks
Point 6: The authors don’t explain the relevance of studying the three cell lines. As clinicians, I think their expertise in this specific cancer is highly valuable, so please explain why you are comparing HuLM to LMS cells. What were you expecting using this cell line in addition to cells from normal tissue? Also in a minor point, the acronyms for these cell lines are not in the main text, only in the Material and methods…
Response 6: Thanks for the comments. We have included the explanation why we were using three kinds of cell lines in the manuscript (Page 16-17, lines 330-333) and have the acronyms for these cell lines throughout the whole manuscript (Page 5, line 99-102).
Thank you, but I think it would help the reader to understand what the different cell types are used for before the result section as well. This way when one dives into the manuscripts one can readily interpret the result of your experiments
Response 6. Thanks for the comment, we have introduced the explanation why we used these cell lines in the methodology part (Page 2, line 79-81).
Point 7: Some conclusions seem to be a little biased. For example, about the repressor drug used in fig S3 and S4 the text states that the decrease in proliferation is more potent in LMS celle compared to UTSM, when the dose-reponse graphs for LDE225 and GDC0449 look very similar for the two cell lines.
Response7: Thanks for the comments. To better compare the inhibitory effect of HH inhibitors in three cell lines we tested, first we regenerated the curve graphs instead of bar figures to better present the dose and time dependent inhibitory effect using HH inhibitors. Secondly we generated new Figures (Figure S4 E and F), which clearly showed that HH inhibitors exhibited a dominantly inhibitor effect on LMS. We have included the new Figure 4E, F as well as corresponding write-up in the manuscript (Page11, Lines 220-225).
This new representation is much clearer. Maybe you should try to find a way to remove the gap between the Y axis and the 24h timepoints, as the reader who is not careful might not have seen that what is displayed is a relative measurement (proliferation) and then wonder where is the T0 timepoint..
Response: Thanks for the comment. The gap between the Y axis and the 24 h timepoints has been removed.
Point 8: Figure 2. I find it doubtful that all the Hh actors considered in the panels A and B would all decrease in the same range and at the latest time point (72h) when treated with LDE225. Are you sure your cells remain healthy with 72h of treatment with that drug? Could you perform a cell toxicity assay (LDH??) ?
Response 8: Thanks very much for the comments. Although the decrease in RNA expression of HH components occurred after 72 hs treatment with HH inhibitor (LDE225), decreased protein levels of HH components such as SMO and GLI1 were observed after treatment with LDE225 for only 24 hs, suggesting that LDE225 treatment altered the HH components more dominantly at protein levels. This is consistent with previous report that LDE225 interacts with SMO in the drug-binding pocket, where it acts as an antagonist, preventing downstream activation of HH signaling (Jain S et al, 2017, OncoTargets and Therapy). In addition, as shown in Figure S8, LDE225 treated LMS did not show cell morphology change as compared to the untreated cells at three time points ( 24 hs, 48 hs, and 72 hs), suggesting that LDE225 treatment alters the LMS phenotype not via cell toxicity. Per reviewer’s comment, we emphasized our statement and included it in the manuscript ( page 12, lines 232-233, page 18, lines 368-371).
OK
Response: Thanks
Point 9: I do not understand why the values for your controls are getting higher with time in panels G and H? It is not the same thing to have a control staying at the same level and see a decrease of 4
the treated condition and to see the controls getting higher without explanation when the treated conditions remain constant, especially if after 24h of treatment there is no visible difference.
Response 9: The graphics show data of cell proliferation analyses and they were created using the absorbance values at 570 nm from all the viable cells. It was expected that untreated cells (control group) showed increasing values over time because they kept growing.
OK, then for clarity’s sake, please normalize to control.
Response: Thanks for the comment. The graphics have been normalized to control (Figure 2 G and H).
Point 10: To conclude, despite the great amount of work behind this study, I think the overall issue is that the scientific question the authors are trying to answer is unclear. If the goal was to assess the role of the Hh pathway in these cells, I would advise to review the data with a Hh specialist to try to make sense of what has been done so far and design better experiments to complete the study (Gli3R, Primary cilium, better interpretation of the data…)
Response 10: Thanks for the comments. In this manuscript, we focused on determining the effects of intervention strategies, including targeting HH pathway and DNA methylation on HH components expression and phenotype in LMS. Further studies will be performed to better clarify the role of this pathway in LMS using in vivo model.
Point 11: If the aim of the study is to get a feel of the Hh pathway as a therapeutic target in this specific cancer, I would advise to go deeper in the characterization of the phenotype associated with the diverse drugs by using more sensitive techniques to add to the data (KI67or BrDU instead of MTT, 3D migration instead of wound healing, Cell cycle analysis, maybe grafted animals…)
Response 11: Thanks for the comments. Although we have several important findings in the manuscript as illustrated in 6 figures and 8 supplemental figures, we agreed with reviewer’s comments that further studies are needed, such as animal work, new and alternative approaches etc., to better understand the role of HH pathway in LMS. We will design all related experiments and hopefully generate exciting data for future manuscripts.
I think the overall quality of the manuscript has increase substantially and the goal is clearer. The main point remaining to solve is what happens with your Gli3 antibody and Gli3R. It is a major issue with the paper, since you’re using Gli3 as a readout of the activation of the pathway and drug activity in almost all of your experiments…
Response: Thanks very much for this important comment. We have addressed this issue on response 2 above
Reviewer 2 Report
The authors rewrote the manuscript and it is much better now. Some of the mistakes are not corrected, mostly in figures. Figures 1C and 3D are the same, and now I also saw that in Figure 3A in quantification of control migration is written ‘Controle’.
Also the Results section is still wrong. It should be 3. RESULTS, and not . 33.RESULTS
Author Response
The authors rewrote the manuscript and it is much better now. Some of the mistakes are not corrected, mostly in figures. Figures 1C and 3D are the same, and now I also saw that in Figure 3A in quantification of control migration is written ‘Controle’.
Also the Results section is still wrong. It should be 3. RESULTS, and not . 33.RESULTS
Response: Thanks for the comments. We performed the correction in the figures and the Results (Page 4, line 155).
Round 2
Reviewer 1 Report
I am still puzzeled by the result about Gli3R, but I see no reason to doubt it and hence I leave it to the reader to make its mind about this. Hopefully new studies will enlight us on this point.
Other than that, manuscript has improved substancially.
This manuscript is a resubmission of an earlier submission. The following is a list of the peer review reports and author responses from that submission.
Round 1
Reviewer 1 Report
The authors have showed that deactivation of SMO, GLI and DNMTs were able to inhibit Uterine leiomyosarcoma malignancy, and the combination of those treatments exhibited a potentiated effect on LMS malignant, leading to a deactivating HH pathway. The authors have concluded that all these data open new perspectives for uterine management, focusing on the development of novel non-invasive specific therapeutics for this aggressive tumor.
Introduction: well written.
Materials and methods: well written.
Results. well written. Statistical analysis was performed and the data is reliable.
Discussion. well written.
Do you have in vivo data? If so, add the histological data as it will improve the quality of your manuscript.
Reviewer 2 Report
The authors present a study of Hedgehog signaling in uterine leiomyosarcoma (LMS). They also assessed the effect of several pathway inhibitors as well as DNA methyltransferase inhibitors (DNMTi) on LMS cells. They found that SMO and GLI inhibitors decreased proliferation of LMS cells and that this effect was increased with DNMTi treatment.
Although the article is interesting and scientifically sound it does not look completely finished and is filled with a lot of small mistakes the authors should have removed themselves.
lines 49-61 – These two sections could be combined in one, or the first one could be about methylation studies about the HH signaling and the second concentrate on PTCH1, as the most studied gene.
line 83 – any mention of GDC0449 is missing. Only later in the results the concentrations are mentioned in a Figure 3S B. Same with Gant58.
line 95 – link is in a larger font than the rest of the text
line 97 – B2M is not in italic
line 104 – city and state is missing for Thermo Fisher
line 109 – city and state is missing for Bio-Rad
line 116 – for Image J software the version and reference is needed
line 123 – city and state is missing for Beckman Coulter Gallios
line 143 – It should be 3. RESULTS
Figure 1C – GLI1, GLI2 and GLI3 are written with spaces
Figure 1B and C – in figure 1B the highest expression of both GLI2 and GLI3 are in HuLM, but in figure 1C the highest expression of both are in nucleus of LMS. The authors should explain this.
lines 173-174 – ‘While Gant58 did not show an effect on proliferation, Based on MTT results…’ – there is a part of the text missing
Figure 3D – The gene legends are not spaced correctly
lines 244, 246, 247, 373, 374 and 377 – PTCH1 should be in italic
line 251 – DNMTs should be in italic
lines 257-259 – The authors claim that the RNA and protein expression of DNTSs was decreased. It is not very clear from the figures and it was not stated if the decrease was significant.
lines 262 and 296 – ‘and’ should not be in italic
line 265 – it should be after 24, 48 and 72 hours, not only 72 hours after treatment in both C and D
line 277 – the sentence ‘Next, determine the effect of DNMT inhibition on proliferation, migration, and apoptosis in LMS cells.’ makes no sense
lines 326 and 351 – it should be GLI1
Reviewer 3 Report
In the present manuscript, entitled "Targeting Hedgehog pathway and DNA methyltransferases in uterine leiomyosarcoma cells ", Natalia Garcia and colleagues propose to study the Hedgehog pathway in a very aggressive kind of tumor, leiomyosarcoma.
Major issues :
- To me, characterization of the Hedeghog pathway in these cells is unclear to say the least, for several reasons:
-The authors present the results as if up/downregulation of any of the Hh actors they choose to study were equivalent and they never discuss the data in detail neither in the result nor in the discussion section. For example, in Figure 1 A (but this holds true throughout the article), Smoothened, is an activator of the downstream elements of the pathway, when SUFU is a repressor. Gli1 is a transcription factor which is an activator of the target genes, but only when the pathway is activated, when Gli2 and Gli3 are not upregulated by the activation of the pathway. Gli2 is the major activator transcription factor, when Gli3 is the major repressor in it’s truncated form…. For instance, can the authors explain why an activator like Smo is upregulated, but also a repressor like SUFU?
-In the same line of thoughts, the authors study Gli3 expression by qPCR, which is not sufficient to decipher between its weak role as an activator and its far more potent role as a repressor since the difference is due to its proteolysis. Then, they perfom WBs which is a good way to check both the full length form (activator) and shorter form (repressor) and still, they fail at showing the repressor from (I’m guessing based on the molecular weight that what they show is the activator form, but it is not even stated…). Although I would agree that the pathway seems constitutively activated given the data for Smo and Gli1, but it’s hard to draw a definitive conclusion when the repressor part is occulted.
-The authors show that the pathway is activated in these cells, but are exogenous activators able to crank up the activation or is the pathway already at full speed? There is no experiment showing what Hh activators (Shh, SAG, purmorphamine) do on these cells.
-I find it quite puzzling that in 2020, nearly 20 years after the discovery of the regulation of the Hedgehog pathway by the primary cilium, researcher would consider publishing a paper without even mentioning the cilium. Just a few questions that I feel should be answered if the aim of the study is really to understand the regulation of Hh in these cells : Are the cells even ciliated? If yes, is Smo in the cilium at the basal state? What about patched does it leave the cilium upon actiavtion? Are Glis properly localized in the cilium?
- Since the characterization of the pathway in its basal state is weak it is very hard to draw conclusions when modulators are added, let alone try to make sense of DNA methylation profiles.
- The authors don’t explain the relevance of studying the three cell lines. As clinicians, I think their expertise in this specific cancer is highly valuable, so please explain why you are comparing HuLM to LMS cells. What were you expecting using this cell line in addition to cells from normal tissue? Also in a minor point, the acronyms for these cell lines are not in the main text, only in the Material and methods…
- Some conclusions seems to be a little biased. For example, about the repressor drug used in fig S3 and S4 the text states that the decrease in proliferation is more potent in LMS celle compared to UTSM, when the dose-reponse graphs for LDE225 and GDC0449 look very similar for the two cell lines.
- Figure 2. I find it doubtful that all the Hh actors considered in the panels A and B would all decrease in the same range and at the latest time point (72h) when treated with LDE225. Are you sure your cells remain healthy with 72h of treatment with that drug? Could you perform a cell toxicity assay (LDH??) ?
- I do not understand why the values for your controls are getting higher with time in panels G and H? It is not the same thing to have a control staying at the same level and see a decrease of the treated condition and to see the controls getting higher without explanation when the treated conditions remain constant, especially if after 24h of treatment there is no visible difference.
To conclude, despite the great amount of work behind this study, I think the overall issue is that the scientific question the authors are trying to answer is unclear. If the goal was to assess the role of the Hh pathway in these cells, I would advise to review the data with a Hh specialist to try to make sense of what has been done so far and design better experiments to complete the study (Gli3R, Primary cilium, better interpretation of the data…)
If the aim of the study is to get a feel of the Hh pathway as a therapeutic target in this specific cancer, I would advise to go deeper in the characterization of the phenotype associated with the diverse drugs by using more sensitive techniques to add to the data (KI67or BrDU instead of MTT, 3D migration instead of wound healing, Cell cycle analysis, maybe grafted animals…)